# Purified EDEM3 or EDEM1 alone produces determinant oligosaccharide structures from M8B in mammalian glycoprotein ERAD

Ginto George[1†‡], Satoshi Ninagawa[1†§], Hirokazu Yagi[2], Jun-ichi Furukawa[3], Noritaka Hashii[4], Akiko Ishii-Watabe[4], Ying Deng[1], Kazutoshi Matsushita[1], Tokiro Ishikawa[1], Yugoviandi P Mamahit[5], Yuta Maki[5,6], Yasuhiro Kajihara[5,6], Koichi Kato[2,7], Tetsuya Okada[1]*, Kazutoshi Mori[1]*

[1]Department of Biophysics, Graduate School of Science, Kyoto University, Kyoto, Japan; [2]Graduate School of Pharmaceutical Sciences, Nagoya City University, Nagoya, Japan; [3]Department of Advanced Clinical Glycobiology, Graduate School of Medicine, Hokkaido University, Sapporo, Japan; [4]Division of Biological Chemistry and Biologicals, National Institute of Health Sciences, Kawasaki, Japan; [5]Department of Chemistry, Graduate School of Science, Osaka University, Toyonaka, Japan; [6]Project Research Center for Fundamental Sciences, Graduate School of Science, Osaka University, Toyonaka, Japan; [7]Exploratory Research Center on Life and Living Systems (ExCELLS) and Institute for Molecular Science, National Institutes of Natural Sciences, Okazaki, Japan

*For correspondence:
tokada@upr.biophys.kyoto-u.ac.jp (TO);
mori@upr.biophys.kyoto-u.ac.jp (KM)

†These authors contributed equally to this work

Present address: ‡Cambridge Institute of Medical Research, University of Cambridge, Cambridge, United Kingdom; §Biosignal Research Center, Kobe University, Kobe, Japan

Competing interest: The authors declare that no competing interests exist.

**ABSTRACT** Sequential mannose trimming of *N*-glycan, from M9 to M8B and then to oligosaccharides exposing the α1,6-linked mannosyl residue (M7A, M6, and M5), facilitates endoplasmic reticulum-associated degradation of misfolded glycoproteins (gpERAD). We previously showed that EDEM2 stably disulfide-bonded to the thioredoxin domain-containing protein TXNDC11 is responsible for the first step (George et al., 2020). Here, we show that EDEM3 and EDEM1 are responsible for the second step. Incubation of pyridylamine-labeled M8B with purified EDEM3 alone produced M7 (M7A and M7C), M6, and M5. EDEM1 showed a similar tendency, although much lower amounts of M6 and M5 were produced. Thus, EDEM3 is a major α1,2-mannosidase for the second step from M8B. Both EDEM3 and EDEM1 trimmed M8B from a glycoprotein efficiently. Our confirmation of the Golgi localization of MAN1B indicates that no other α1,2-mannosidase is required for gpERAD. Accordingly, we have established the entire route of oligosaccharide processing and the enzymes responsible.

## Editor's evaluation

This study demonstrates the details of mannose trimming in the ER by EDEM1 and EDEM3, filling an important gap in how mannose trimming creates an ERAD signal and shunts glycoproteins to the ERAD pathway.

## Introduction

Eukaryotic cells are equipped with a sophisticated system to handle proteins which are unfolded or misfolded in the endoplasmic reticulum (ER), where newly synthesized secretory and transmembrane

proteins destined for the secretory pathway gain their own three-dimensional structure (*Bukau et al., 2006*). These functionally inappropriate proteins are somehow detected in the ER lumen and then transferred to the retrotranslocational channel embedded within the ER membrane, termed the retrotranslocon, followed by ubiquitin-dependent degradation by the proteasome in the cytoplasm. This series of processes is collectively referred to as ER-associated degradation (ERAD-L; L for lumen) (*Ninagawa et al., 2021*).

Extensive analysis of the mechanism of glycoprotein ERAD-L (gpERAD) has revealed that particular structures of the carbohydrate moiety become a signal for degradation (*Ninagawa et al., 2021*). An oligosaccharide of 14 sugar units, consisting of 3 glucose, 9 mannose, and 2 *N*-acetyl-glucosamine molecules ($Glc_3Man_9GlcNAc_2$, abbreviated as G3M9 hereafter), is transferred to an Asn residue in the consensus sequence (Asn-X-Ser/Thr; X: any amino acid except Pro) present in newly synthesized protein in yeast and mammalian cells. G3M9 is converted to M9 by the sequential actions of Gls1 and Gls2-Gtb1 in yeast and glucosidases I and II in mammalian cells. Productive folding of the protein moiety is facilitated during this period by the calnexin/calreticulin cycle in mammalian cells, which relies on G1M9-specific lectin-type chaperones (calnexin and calreticulin) associated with the oxidoreductase ERp57, and UDP-glucose:glycoprotein glucosyltransferases 1 and 2, which are capable of re-adding glucose to M9 if the protein moiety is not yet folded. It should be noted that the Cne1 (an orthologue of calnexin and calreticulin) cycle is not effective in yeast because a functional orthologue of UGGTs is not present in yeast (*Saccharomyces cerevisiae*). After completion of folding, the glycoprotein is transported to the next compartment in the secretory pathway, the Golgi apparatus. However, if the protein moiety is not folded within a certain threshold time, the glycoprotein undergoes gpERAD via the conversion of M9 to M8B and then to oligosaccharides exposing the α1,6-linked mannosyl residue, namely M7A, M6, and M5. α1,6-Linked mannosyl residue-specific lectins OS9 and XTP3B transfer the glycoprotein to the retrotranslocon for subsequent degradation in the cytoplasm in mammalian cells (*Ninagawa et al., 2021*).

In yeast, the α1,2-mannosidases Mns1 and Htm1 carry out the first and second mannose trimming steps, respectively (*Ninagawa et al., 2021*). We have constructed knockout (KO) cells for α1,2-mannosidase candidate genes, namely EDEM1, EDEM2, and EDEM3 (Htm1 orthologues), in human HCT116 diploid cells, and have shown that EDEM2 is required for the conversion of M9 to M8B and that EDEM3 (mainly) and EDEM1 (partly) are required for the conversion of M8B to oligosaccharides, which exposes the α1,6-linked mannosyl residue (*Ninagawa et al., 2014*). Further, our recent biochemical analysis has clarified that although EDEM2 alone exhibits no mannosidase activity toward M9, as reported previously (*Mast et al., 2005*; *Shenkman et al., 2018*), EDEM2 stably disulfide-bonded to the thioredoxin domain-containing protein TXNDC11 catalyzes the first mannose trimming step from M9 to M8B in vitro (*George et al., 2020*).

Intramolecular disulfide bond formation in the mannosidase homology domain (MHD) is considered essential for EDEM2 to exhibit α1,2-mannosidase activity because mutation of either C65 or C408 inactivated EDEM2 in gpERAD (*George et al., 2020*). Here, we examined whether the disulfide bonding between C65 and C408 is indeed formed only in the presence of TXNDC11. We then examined whether disulfide bond formation in the MHD is essential for EDEM1 or EDEM3 to exhibit α1,2-mannosidase activity, and whether TXNDC11 is disulfide-bonded to EDEM1 or EDEM3. Finally, we examined whether purified EDEM3 or purified EDEM1 can catalyze the second mannose trimming step from M8B to oligosaccharides exposing the α1,6-linked mannosyl residue.

## Results

### Formation of essential disulfide bonding in EDEM2 in the presence of TXNDC11

As we reported previously, analysis by non-reducing SDS-PAGE followed by immunoblotting showed that introduction of 3xFlag-tagged EDEM2 into EDEM2-KO cells produced its monomer, and probably dimer and aggregated forms, whereas simultaneous introduction of 3xFlag-tagged TXNDC11 and 3xFlag-tagged EDEM2 into EDEM2-KO cells produced a high molecular weight complex of EDEM2 and TXNDC11, designated EDEM2[§]–3xFlag (*Figure 1A*). It should be noted

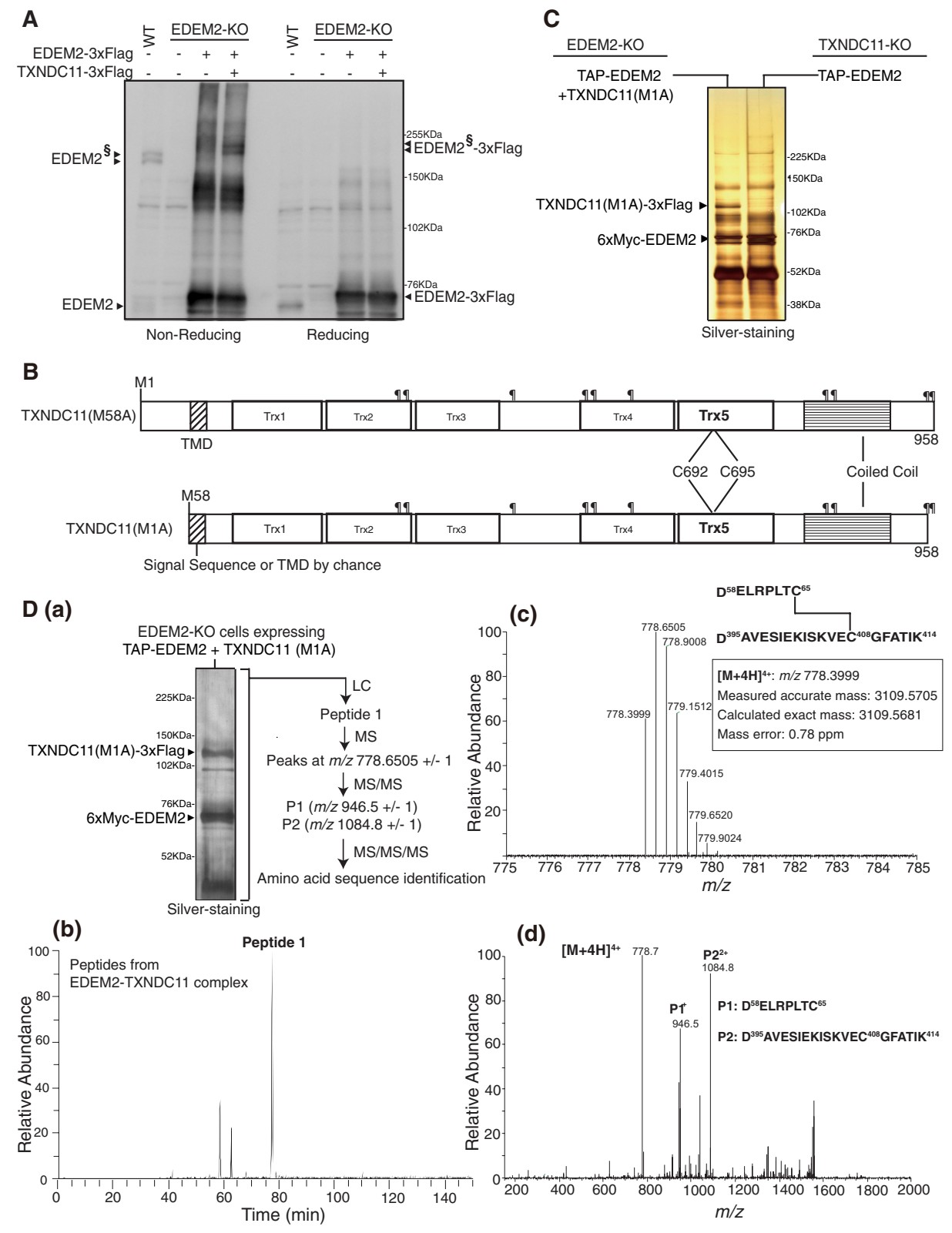

**Figure 1.** Identification of a disulfide-linked peptide in EDEM2-TXNDC11 complex by liquid chromatography (LC)/mass spectrometry (MS). (**A**) Cell lysates were prepared from untransfected wild-type (WT) cells and EDEM2-knockout (KO) cells untransfected or transfected with (+) or without (-) plasmid to express EDEM2-3xFlag or TXNDC11-3xFlag, subjected to SDS-PAGE under non-reducing and reducing conditions, and analyzed by immunoblotting using anti-EDEM2 antibody. EDEM2§ denotes EDEM2 stably disulfide-bonded to TXNDC11. (**B**) Structures of the M58A and M1A

*Figure 1 continued on next page*

Figure 1 continued

mutants of human TXNDC11 containing the transmembrane domain (TMD), five Trx domains, and coiled coil domain are shown schematically. ¶ denotes potential *N*-glycosylation sites. (**C**) Eluates were obtained from EDEM2-KO cells overexpressing TAP-EDEM2 plus TXNDC11(M1A) and from TXNDC11-KO cells overexpressing TAP-EDEM2, subjected to SDS-PAGE under reducing conditions, and silver-stained. The positions of TXNDC11(M1A)–3xFlag and 6xMyc-EDEM2 are indicated. (**D**) (**a**) EDEM2 stably disulfide-bonded to TXNDC11 was purified at a larger scale and silver-stained after reducing SDS-PAGE. The eluate was analyzed sequentially by LC/MS, MS/MS, and MS/MS/MS as indicated. This experiment was conducted once. (**b**) Extracted ion chromatogram of the ion at *m/z* 778.40 (±0.01) from EDEM2-TXNDC11 complex is shown. (**c**) MS spectrum of Peptide 1 observed in (**b**) is shown. The six peaks other than *m/z* 778.3999 are isotopic ($^{13}$C-containing) ion peaks. (**d**) Electron-transfer/higher-energy collisional dissociation-tandem mass spectrometry (EThcD-MS/MS) spectrum of the ion at *m/z* 778.6505 ± 1 (*m/z* 778.3999–779.4015) derived from Peptide 1 (**c**) is shown.

The online version of this article includes the following figure supplement(s) for figure 1:

**Figure supplement 1.** Purification strategy.

**Figure supplement 2.** Higher-energy collisional dissociation (HCD)-mass spectrometry (MS)/MS/MS spectra.

that both TXNDC11 and the EDEM2-TXNDC11 complex are detected as a doublet band due to alternative translational initiation of TXNDC11 at M1 or M58 (*George et al., 2020*).

To determine disulfide bonding status in the MHD, we intended to purify noncomplexed EDEM2 and EDEM2-TXNDC11 complex separately and subject them to liquid chromatography (LC)/mass spectrometry (MS) analysis. For this purpose, TXNDC11-KO cells were transfected with plasmid to express tandem affinity purification (TAP)-tagged EDEM2 to purify noncomplexed EDEM2; this TAP consists of 2 × immunoglobulin G-binding domain of protein A, 2 × TEV protease recognition site, and 6xMyc (*Figure 1—figure supplement 1A*). Also, EDEM2-KO cells were transfected with plasmid to express TAP-tagged EDEM2 plus plasmid to express TXNDC11(M1A). It should be noted that TXNDC11(M1A) was used to purify EDEM2-TXNDC11 complex; TXNDC11(M58A) is expressed only as a transmembrane protein, whereas TXNDC11(M1A) is expressed as both soluble and transmembrane proteins because its N-terminal hydrophobic region functions by chance as either a signal sequence or a transmembrane domain (TMD) (*Figure 1B*), allowing us to purify a soluble complex of EDEM2 and TXNDC11.

6xMyc-tagged EDEM2 noncomplexed or complexed with TXNDC11(M1A) was purified as depicted in *Figure 1—figure supplement 1B*, and silver-staining showed expected bands (*Figure 1C*). These proteins purified at a larger scale (*Figure 1D (a)* ) were digested with Asp-N and the resulting peptides were analyzed by LC/MS. The results of peptides derived from EDEM2-TXNDC11 complex showed that 'Peptide 1' was eluted at 77.38 min in LC (*Figure 1D (b)*) and produced an ion peak at *m/z* 778.3999, which contained only $^{12}$C as C, plus six other isotopic ($^{13}$C-containing) ion peaks in MS (*Figure 1D (c)*). This 'Peptide 1' was not detected in peptides derived from noncomplexed EDEM2 (data not shown). Importantly, the measured accurate mass of the ion peak at *m/z* 778.3999 matched the calculated exact mass of the two peptides (P1: D58~C65 of EDEM2 and P2: D395~K414 of EDEM2) covalently connected by disulfide bonding between C65 and C408 of EDEM2 (*Figure 1D (c)*). Indeed, subsequent electron-transfer/higher-energy collisional dissociation (EThcD)-MS/MS analysis of Peptide 1 (*m/z* 778.6505 ± 1) produced P1 at *m/z* 946.5 and P2 at *m/z* 1084.8 by preferential cleavage of disulfide bonds (*Figure 1D (d)*). Further, higher-energy collisional dissociation (HCD)-MS/MS/MS analysis of P1 (*m/z* 946.5 ± 1) and P2 (*m/z* 1084.8 ± 1), during which peptide bonds are cleaved, confirmed that P1 corresponded to D[58]ELRPLTC[65], whereas P2 corresponded to D[395]AVESIEKISKVEC[408]GFATIK[414] (*Figure 1—figure supplement 2*). We concluded that TXNDC11 helps EDEM2 to form the disulfide bond between C65 and C408, which is essential for EDEM2 to exhibit α1,2-mannosidase activity (*George et al., 2020*).

## Characterization of EDEM1 and EDEM3

C65 of human EDEM2 is conserved as C160 of human EDEM1 and C82 of human EDEM3, whereas C408 of human EDEM2 is conserved as C529 of human EDEM1 and C441 of human EDEM3 (*Figure 2A*). To determine their importance, we mutated all cysteine residues present in EDEM1 and EDEM3. To evaluate their functionality in gpERAD, we constructed EDEM1, 3-double KO (DKO) cells (two independent clones #1 and #2, *Figure 2—figure supplement 1A* and S1B), which expressed neither EDEM1 mRNA nor EDEM3 mRNA (*Figure 2B*), and grew slightly more slowly

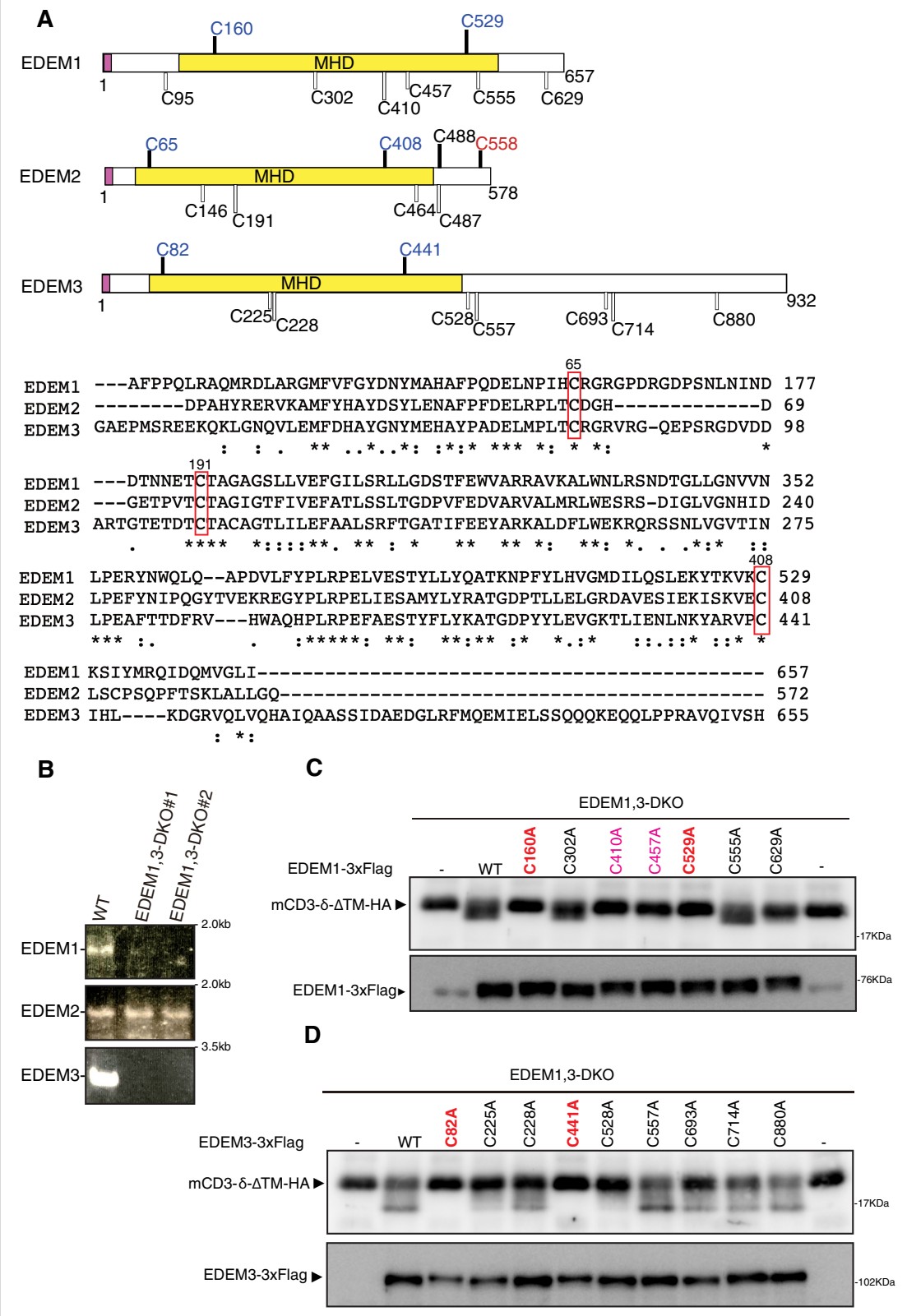

**Figure 2.** Effect of mutation of various cysteine residues in EDEM1 and EDEM3 on endoplasmic reticulum-associated degradation of misfolded glycoproteins (gpERAD). (**A**) Structures of human EDEM1, EDEM2, and EDEM3 are schematically shown with cysteine residues (C) highlighted together with their positions (black bars underneath C indicate conserved cysteine residues, whereas white bars over C indicate non-conserved cysteine residues). The purple and yellow boxes denote the signal sequence and mannosidase homology domain (MHD), respectively. Sequence comparison around the

*Figure 2 continued on next page*

*Figure 2 continued*

three cysteine residues (C65, C191, and C408 of EDEM2) is shown below (asterisk and colon indicate identical and similar amino acids, respectively). (**B**) RT-PCR to amplify cDNA corresponding to full-length open reading frame in EDEM1/2/3 mRNA in wild-type (WT) and EDEM1, 3-double knockout (DKO) cells (two independent clones #1 and #2) is shown. (**C**) Cell lysates were prepared from EDEM1, 3-DKO cells expressing WT or one of various cysteine mutants of 3 × Flag-tagged EDEM1 together with mCD3-δ-ΔTM-HA by transfection, and analyzed by immunoblotting using anti-HA and anti-EDEM1 antibodies. (**D**) Cell lysates were prepared from EDEM1, 3-DKO cells expressing WT or one of various cysteine mutants of 3 × Flag-tagged EDEM3 together with mCD3-δ-ΔTM-HA by transfection, and analyzed by immunoblotting using anti-HA and anti-Flag antibodies.

The online version of this article includes the following figure supplement(s) for figure 2:

**Figure supplement 1.** Construction and characterization of EDEM1, 3-double knockout (3-DKO) cells.

than wild-type (WT) cells (*Figure 2—figure supplement 1C*), as we previously described for the construction of EDEM1/2/3-triple KO (TKO) cells (*Ninagawa et al., 2015*). We used #1 as EDEM1, 3-DKO cells hereafter. M9 was accumulated in EDEM-TKO cells as we described previously (*Ninagawa et al., 2015*), whereas M8B was accumulated in EDEM1, 3-DKO cells (*Figure 2—figure supplement 1D*), as we expected. Accordingly, degradation of human ATF6α, a transmembrane-type gpERAD substrate, and mCD3-δ-ΔTM-HA, a soluble gpERAD substrate, was markedly delayed in EDEM1, 3-DKO cells (*Figure 2—figure supplement 1E* and 1F), similarly to the case of EDEM2-KO cells (*Ninagawa et al., 2014*).

We employed a gpERAD substrate migration assay. mCD3-δ-ΔTM-HA, possessing three *N*-glycosylation sites, migrates slightly faster due to a mannose trimming-mediated decrease in its molecular weight if cotransfected α1,2-mannosidase is active in gpERAD, and migration position is closely correlated with degradation rate (*George et al., 2020*). We describe this effect on migration as a downward shift hereafter. Results showed that the mutations C160A and C529A of EDEM1 as well as C82A and C441A of EDEM3 indeed inactivated EDEM1 and EDEM3, respectively, in gpERAD (*Figure 2C and D*). Thus, the conserved cysteine residues, at which intramolecular disulfide bonding was confirmed in EDEM2, turned out to be critical for EDEM1 and EDEM3 to exhibit α1,2-mannosidase activities, as expected. Results also showed that the mutations C410A and C457A of EDEM1 inactivated the α1,2-mannosidase activity of EDEM1 (*Figure 2C*), suggesting that C410 and C457 of EDEM1 might form a local disulfide bond to stabilize the MHD of EDEM1. In this connection, it was previously suggested that four of eight cysteine residues present in EDEM1 can act as free thiols and may be involved in thiol-dependent interaction with substrates, whereas the other four cysteine residues are in sufficient proximity to form two disulfide bonds (C160-C529 and C410-C457) (*Lamriben et al., 2018*). In contrast, cysteine residues outside of the MHD of EDEM1 and EDEM3 are not important for their α1,2-mannosidase activity (*Figure 2C and D*), unlike C558 of EDEM2, which is stably disulfide-bonded to C692 of TXNDC11 (*George et al., 2020*).

## Subcellular localization of MAN1B1

Our previous genetic analysis revealed that the α1,2-mannosidase MAN1B1 (a sole Mns1 orthologue) appears to play only a very minor role in gpERAD (*Ninagawa et al., 2014*), contrary to the original proposals (*Avezov et al., 2008*; *Gonzalez et al., 1999*; *Hosokawa et al., 2003*). To further evaluate the contribution of MAN1B1 to the mannose trimming step in gpERAD, we determined the subcellular localization of MAN1B1 fused to mCherry which was expressed by transfection in HCT116 cells or HeLa cells under the control of the CMV promoter or its truncated version, which we termed CMVshort promoter; the CMVshort promoter is significantly weaker than the CMV promoter in driving gene expression (*Nadanaka et al., 2004*).

MAN1B1-mCherry driven by the CMV promoter or CMVshort promoter was not colocalized with mEGFP-KDEL, an ER marker, but was colocalized with mEGFP-Giantin, a Golgi marker, in HCT116 cells (*Figure 3*). Because signaling from Lamp1-mGFP, a marker of lysosome, was weak in HCT116 cells, we also checked their colocalization in HeLa cells. Again, MAN1B1-mCherry driven by the CMV promoter or CMVshort promoter was not colocalized with mEGFP-KDEL but rather with mEGFP-Giantin in HeLa cells (*Figure 3—figure supplement 1*). Interestingly, some MAN1B1-mCherry highly expressed from the CMV promoter overlapped with Lamp1-mGFP (*Figure 3—figure supplement 1F*), whereas MAN1B1-mCherry never overlapped with mEGFP-KDEL, even after higher expression

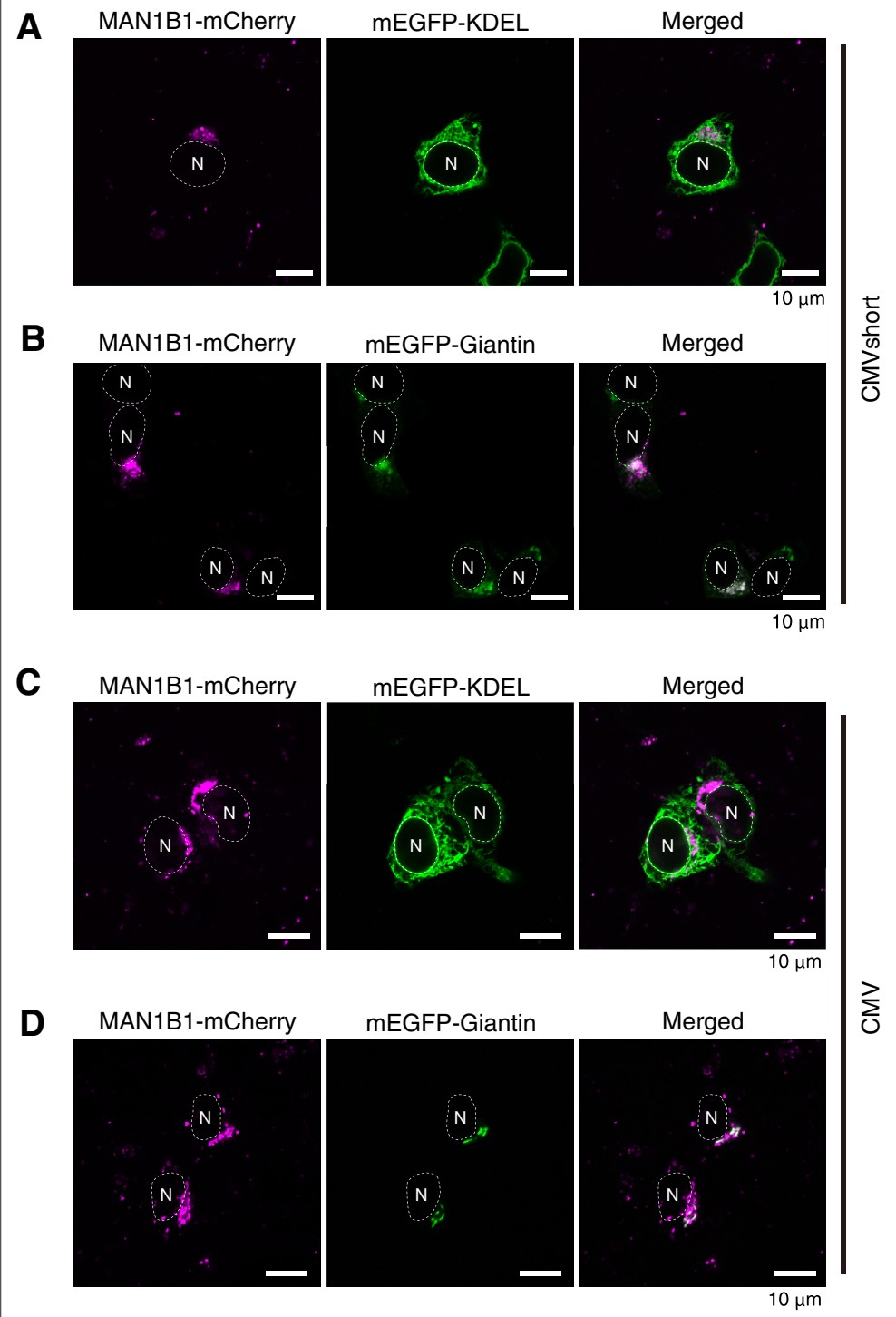

**Figure 3.** Localization of MAN1B1-mCherry in HCT116 cells. (**A**, **B**) HCT116 cells were transfected with plasmid to express MAN1B1-mCherry under the control of the CMVshort promoter together with plasmid to express mEGFP-KDEL (**A**) or mEGFP-Giantin (**B**), and analyzed by confocal microscopy (Airyscan). Scale bar: 10 µm. (**C, D**) HCT116 cells were transfected with plasmid to express MAN1B1-mCherry under the control of the CMV promoter together with plasmid to express mEGFP-KDEL (**C**) or mEGFP-Giantin (**D**), and analyzed by confocal microscopy (Airyscan). Sale bar: 10 µm.

The online version of this article includes the following figure supplement(s) for figure 3:

**Figure supplement 1.** Localization of MAN1B1-mCherry in HeLa cells.

(*Figure 3—figure supplement 1D*). Thus, MAN1B1-mediated mannose trimming cannot participate in gpERAD, the events in the ER.

## Purification of EDEM3 and EDEM1

We next examined whether TXNDC11 is disulfide-bonded to EDEM1 or EDEM3. Simultaneous introduction of 3xFlag-tagged TXNDC11 and 3xFlag-tagged EDEM1 into EDEM1-KO cells or simultaneous introduction of 3xFlag-tagged TXNDC11 and 3xFlag-tagged EDEM3 into EDEM3-KO cells did not change band patterns compared with single introduction of 3xFlag-tagged EDEM1 into EDEM1-KO cells or single introduction of 3xFlag-tagged EDEM3 into EDEM3-KO cells (*Figure 4A* (a, b)). When any of 3xFlag-tagged EDEM1, EDEM2, and EDEM3 was introduced into WT HCT116 cells, followed by immunoprecipitation with anti-Flag antibody and subsequent immunoblotting using anti-TXNDC11 antibody under reducing and non-reducing conditions, endogenous TXNDC11 was clearly co-immunoprecipitated from cells overexpressing 3xFlag-tagged EDEM2, indicating that EDEM2 is the main partner of TXNDC11 (*Figure 4B*). Accordingly, we intended to purify noncomplexed EDEM1 and EDEM3 from HCT116 cells overexpressing respective TAP-tagged protein.

We employed MAN1B1 as a control of an extensive α1,2-mannosidase, based on a report that it can produce M6 and M5 from M9 in vitro (*Aikawa et al., 2012*); we confirmed this observation (see *Figure 5C*). Because MAN1B1 is a transmembrane protein, we deleted its TMD-containing N-terminal 105 aa to obtain MAN1B1(Δ105) for efficient purification (*Figure 4C (a)*). Both TAP-tagged MAN1B1 and MAN1B1(Δ105) are active mannosidases, as their overexpression in EDEM2-KO cells caused a downward shift to the transfected mCD3-δ-ΔTM-HA band (*Figure 4C (b)*).

Our initial purification of 6xMyc-tagged EDEM1, EDEM3, and MAN1B1(Δ105) from HCT116 cells overexpressing TAP-tagged EDEM1, EDEM3, and MAN1B1(Δ105), respectively, followed by silver staining and immunoblotting, revealed that 6xMyc-tagged EDEM1 was purified as a single band, whereas 6xMyc-tagged EDEM3 was purified as two bands designated EDEM3 and EDEM3[C] (*Figure 1—figure supplement 1C*, c for cleaved). Sequence search revealed one internal TEV protease recognition site in EDEM3 (*Figure 4D (a)*), and the mutation Q543N was sufficient to render EDEM3 resistant to digestion with TEV protease (*Figure 4D (b)*).

## EDEM3 and EDEM1 produce oligosaccharides exposing the α1,6-linked mannosyl residue from pyridylamine-labeled M8B

As M9 is accumulated in EDEM2-KO cells (*Ninagawa et al., 2014*) and M8B is accumulated in EDEM1, 3-DKO cells (*Figure 2—figure supplement 1D*), transfected mCD3-δ-ΔTM-HA migrated slightly more slowly in EDEM2-KO cells than in WT cells (*Figure 5A*, compare lane 1 with lane 2) or marginally more slowly in EDEM1, 3-DKO cells than in WT cells (*Figure 5A*, compare lane 7 with lane 8). When transfected into EDEM2-KO cells, only TAP-tagged EDEM2 caused a downward shift to the transfected mCD3-δ-ΔTM-HA band among TAP-tagged EDEM1, EDEM2, and EDEM3(Q543N) (*Figure 5A*. lane 4), whereas TAP-tagged EDEM1 and EDEM3(Q543N) but not TAP-tagged EDEM2 did so when transfected into EDEM1, 3-DKO cells (*Figure 5A*, lanes 9 and 11), consistent with the results of our gene KO analysis, which showed that EDEM2 is required for the conversion of M9 to M8B and that EDEM3 and EDEM1 are required for the conversion of M8B to oligosaccharides exposing the α1,6-linked mannosyl residue (*Ninagawa et al., 2014*).

6xMyc-tagged EDEM1 and EDEM3(Q543N) were purified as a single band on silver staining and immunoblotting in addition to 6xMyc-tagged EDEM2-TXNDC11 complex (*Figure 5B*). After 24 hr incubation with purified 6xMyc-tagged EDEM3(Q543N), pyridylamine (PA)-labeled M8B was converted to M7, M6, and M5 (*Figure 5C*). M7 turned out to be M7A and M7C (*Figure 5D (a)*), indicating no preference of EDEM3 toward the mannose residue present in the A or C branch of M8B. Purified 6xMyc-tagged EDEM1 exhibited similar mannosidase activity toward M8B, albeit with much weaker activity than 6xMyc-tagged EDEM3(Q543N) (*Figure 5C*). In contrast, M8B was converted to M7 only slightly by purified EDEM2-TXNDC11 complex, and M6 and M5 were hardly detected even after 24 hr incubation (*Figure 5C*), indicating a strong preference of EDEM2-TXNDC11 complex toward M9 (*George et al., 2020*). Based on these results, we have established the route of oligosaccharide processing in mammalian gpERAD (*Figure 5E*).

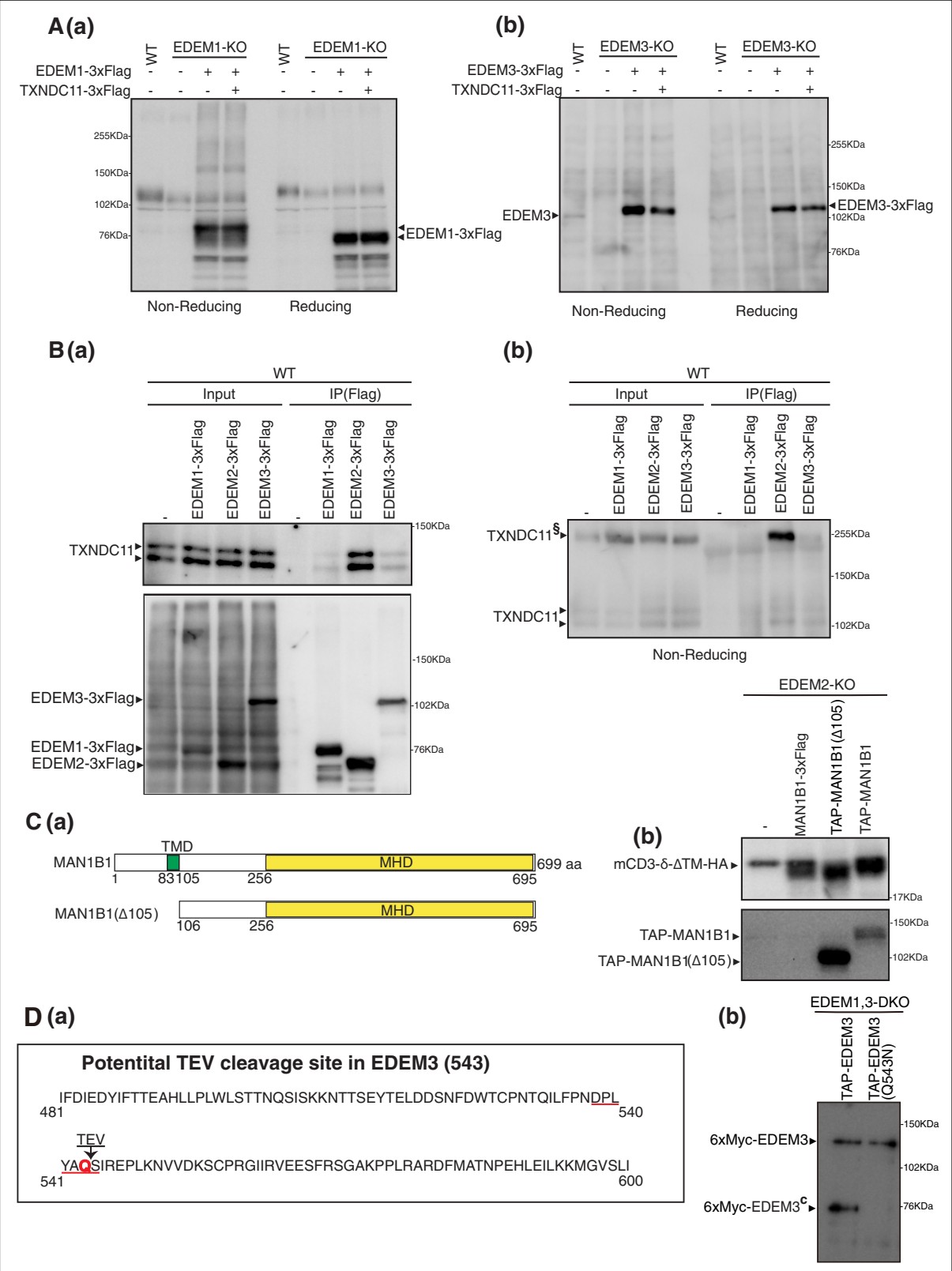

**Figure 4.** Effect of coexpression of TXNDC11 on EDEM1 and EDEM3. (**A**) (**a**) Cell lysates were prepared from untransfected wild-type (WT) cells and EDEM1-knockout (KO) cells untransfected or transfected with (+) or without (-) plasmid to express EDEM1-3xFlag or TXNDC11-3xFlag, subjected to SDS-PAGE under non-reducing and reducing conditions, and analyzed by immunoblotting using anti-EDEM1 antibody. (**b**) Cell lysates were prepared from untransfected WT cells and EDEM3-KO cells untransfected or transfected with (+) or without (-) plasmid to express EDEM3-3xFlag or TXNDC11-

*Figure 4 continued on next page*

Figure 4 continued

3xFlag, subjected to SDS-PAGE under non-reducing and reducing conditions, and analyzed by immunoblotting using anti-EDEM3 antibody. (**B**) Cell lysates were prepared from WT cells untransfected or transfected with plasmid to express EDEM1-3xFlag, EDEM2-3xFlag, or EDEM3-3xFlag, and subjected to immunoprecipitation using anti-Flag antibody. Aliquots of cell lysates (Input) and immunoprecipitates (IP[Flag]) were subjected to SDS-PAGE under reducing (**a**) and non-reducing (**b**) conditions, and analyzed by immunoblotting using anti-TXNDC11 and anti-Flag antibodies. TXNDC11§ denotes TXNDC11 stably disulfide-bonded to EDEM2. (**C**) (**a**) Structures of MAN1B1 and MAN1B1(Δ105) are schematically shown. TMD denotes the transmembrane domain. (**b**) EDEM2-KO cells were transfected with plasmid to express mCD3-δ-ΔTM-HA together with or without plasmid to express MAN1B1-3xFlag, TAP-MAN1B1(Δ105), or TAP-MAN1B1. Cell lysates were then prepared and analyzed by immunoblotting using anti-HA and anti-Myc antibodies. (**D**) (**a**) Location of potential TEV cleavage site in EDEM3 is shown. Its consensus sequence is E-X-X-Y-X-Q-G/S. (**b**) EDEM1, 3-DKO cells expressing TAP-EDEM3 or TAP-EDEM3(Q543N) by transfection were subjected to small-scale purification as in *Figure 1—figure supplement 1B*. Eluates were analyzed by immunoblotting using anti-Myc antibody.

## EDEM3 and EDEM1 produce oligosaccharides exposing the α1,6-linked mannosyl residue from M8B on a protein

We finally examined the mannosidase activity of purified EDEM1, EDEM2-TXNDC11 complex, and EDEM3 toward M8B on a purified gpERAD substrate, such as mCD3-δ-ΔTM-HA (*Figure 6*) and ATF6α(C) (*Figure 7*); ATF6α(C) represents the luminal region of the unfolded protein response transducer ATF6α, whose rapid degradation completely depends on EDEM-mediated mannose trimming (*Horimoto et al., 2013*; *Ninagawa et al., 2014*). 6xMyc-tagged mCD3-δ-ΔTM-HA and ATF6α(C) tagged with 3xMyc were purified from EDEM1, 3-DKO cells overexpressing TAP-tagged mCD3-δ-ΔTM-HA and ATF6(C) tagged with TAP2 (see *Figure 1—figure supplement 1A*), respectively, by transfection (*Figure 6A (a)* and 7 A (a)). They were expected to possess N-glycans of M8B, which we confirmed by MS analysis (see top panel of *Figure 6B (b)* and *Figure 7B (b)*); it should be noted that each oligosaccharide was detected as two peaks consisting of [M + H]⁺, a proton adduct ion, and [M + Na]⁺, a sodium adduct ion. Indeed, they were completely sensitive to digestion with endoglycosidase H (Endo H) (*Figure 6A (b)* and 7 A (b)). Because 6xMyc-tagged mCD3-δ-ΔTM-HA was detected as a doublet band by both anti-Myc and anti-HA antibodies (*Figure 6A (b)*), we consider that the upper and lower bands were produced by cleavage at the first and second TEV cleavage sites, respectively (see *Figure 1—figure supplement 1A*), whereas the doublet band of ATF6α (C) reflected partial degradation during purification.

The band of 6xMyc-tagged mCD3-δ-ΔTM-HA purified from transfected EDEM1, 3-DKO cells exhibited a downward shift after incubation with 6xMyc-tagged EDEM1 (*Figure 6B (a)*, lane 4) and EDEM3(Q543N) (lane 10) but not with 6xMyc-tagged EDEM2-TXNDC11 complex (lane 7). MS analysis of 6xMyc-tagged mCD3-δ-ΔTM-HA samples after 24 hr incubation with 6xMyc-tagged EDEM1 and EDEM3(Q543N) revealed that M8B was converted to M5 almost completely, whereas a major peak was still M8B after 24 hr incubation with 6xMyc-tagged EDEM2-TXNDC11 complex (*Figure 6B (b)*). Quite similar results were obtained with ATF6α(C) tagged with 3xMyc, which was purified from over-expressed EDEM1, 3-DKO cells (*Figure 7A and B*), although somehow a greater amount of M5 was produced after 24 hr incubation of ATF6α(C) with 6xMyc-tagged EDEM2-TXNDC11 complex than was the case with mCD3-δ-ΔTM-HA (*Figure 7B (b)*). Note that the peak of 1742.412 (+NT column, *Figure 6B (b)*) and those of 2054.668 and 1892.627 (+EDEM1 column, *Figure 7B (b)*) were complex-type oligosaccharides derived from immunoglobulin G (silver-stained around a molecular weight marker of 52 kDa, *Figure 6A (a)* and 7 A (a)) possessing only complex-type oligosaccharides, which was partially detached from IgG Sepharose beads. A band silver-stained around a molecular weight marker of 76 kDa (*Figure 6A (a)* and 7 A (a)) was identified to be BiP by MS analysis in our previous study (*Sato et al., 2011*), which possesses no N-glycan, as evidenced by its resistance to digestion with Endo H (*Figure 6A (b)* and 7 A (b)). We concluded that EDEM3 and EDEM1 are more active on protein-bound M8B than on free M8B.

## Discussion

Identification of the exact route of oligosaccharide processing from M9 to M8B and then to oligosaccharides exposing the α1,6-mannosyl residue (M7A, M6, and M5), and of the enzymes involved in it is critical to understanding the molecular mechanism of the mammalian gpERAD. Our previous genetic and biochemical analyses have determined that EDEM2 stably disulfide-bonded

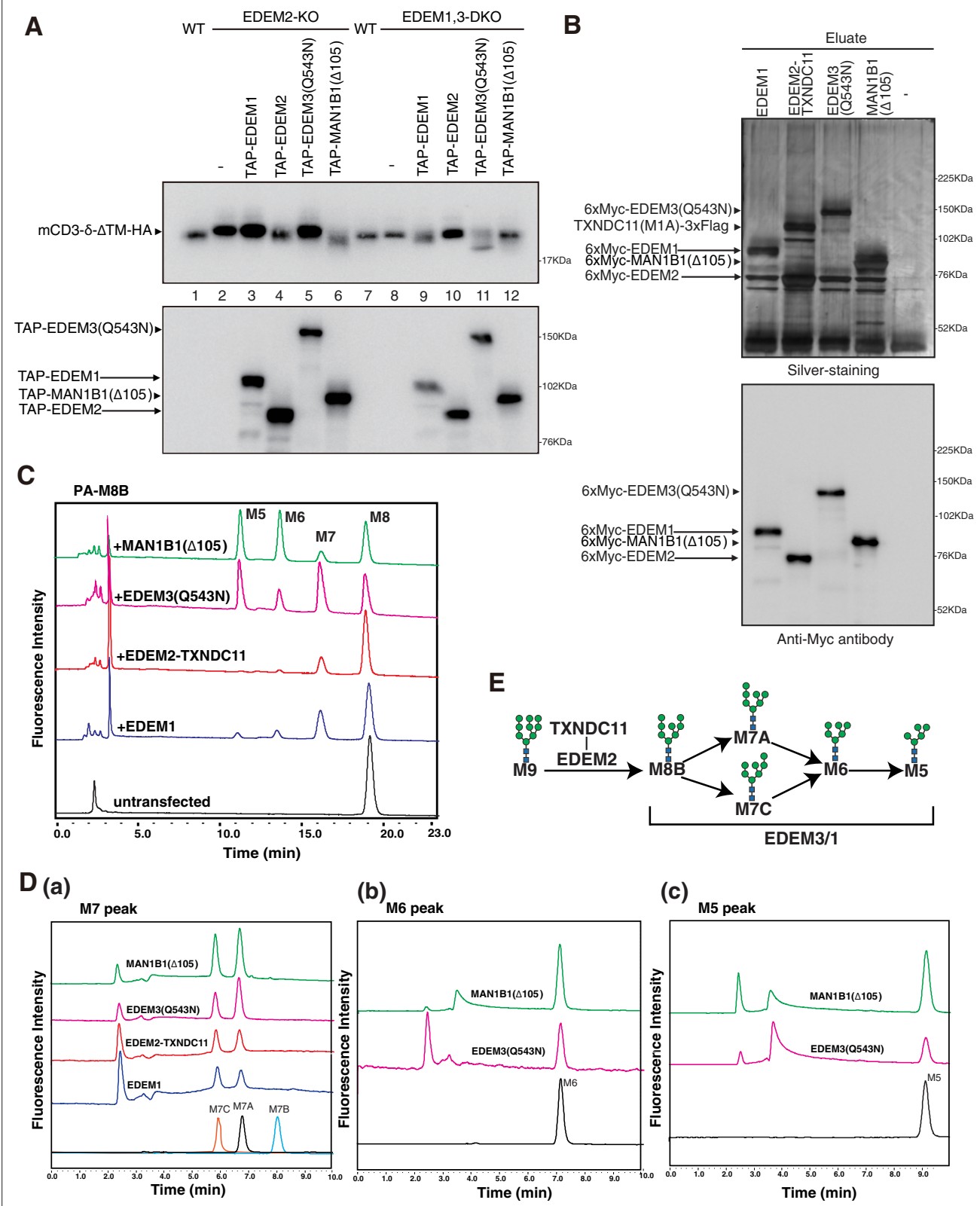

**Figure 5.** Effect of four purified α1,2-mannosidases on PA-M8B. (**A**) Wild-type (WT) cells were transfected with plasmid to express mCD3-δ-ΔTM-HA. EDEM2-knockout (KO) cells and EDEM1, 3-double knockout (DKO) cells were transfected with plasmid to express mCD3-δ-ΔTM-HA together with or without plasmid to express TAP-EDEM1, TAP-EDEM2, TAP-EDEM3(Q543N), or TAP-MAN1B1(Δ105). Cell lysates were then prepared and analyzed by immunoblotting using anti-HA and anti-Myc antibodies. (**B**) Eluates obtained from WT cells untransfected (-) or transfected with plasmid to express TAP-

*Figure 5 continued on next page*

Figure 5 continued

EDEM1, TAP-EDEM2 plus TXNDC11(M1A), EDEM3(Q543N), or MAN1B1(Δ105) were subjected to SDS-PAGE under reducing conditions, silver-stained, and then analyzed by immunoblotting using anti-Myc antibody. PA-M8B was incubated with samples in (**B**) for 24 hr as indicated and then analyzed by HPLC (amide column) for mannose contents. (**D**) The M7 peak (**a**), M6 peak (**b**), and M5 peak (**c**) obtained in (**C**) were analyzed by HPLC (ODS column) for isomer identification. (**E**) Route of oligosaccharide processing in the mammalian endoplasmic reticulum-associated degradation of misfolded glycoproteins (gpERAD) is shown.

to TXNDC11 is responsible for the first mannose trimming step from M9 (*George et al., 2020*; *Ninagawa et al., 2014*). Our biochemical analysis in the present paper together with our previous genetic analysis (*Ninagawa et al., 2014*) has determined that EDEM3 alone (mainly) or EDEM1 alone (partly) is responsible for the second mannose trimming step from M8B.

Our current results with another α1,2-mannosidase MAN1B1 (*Figure 3* and *Figure 3—figure supplement 1*) are well consistent with a previous report showing the Golgi localization of endogenous MAN1B1 by immunofluorescence (*Pan et al., 2011*). Unfortunately, however, we are unable to reconcile our results with findings (*Benyair et al., 2015*) claiming that MAN1B1 is localized in quality control vesicles to which ERAD substrates are transported and in which they interact with the enzyme. Nonetheless, it is not clear at all whether such vesicles contain molecules downstream of the exposure of α1,6-linked mannosyl residue, such as OS9 and XTP3B, as well as the retrotranslocon for efficient degradation of substrates already destined for gpERAD via mannose trimming. The system becomes effective if all components of gpERAD exist in the same compartment, namely in the ER.

Importantly, EDEM3 produced not only M7A but also M6 and M5 from M8B (*Figure 5C*). EDEM1 exhibited a similar tendency, albeit weakly; potential disulfide bonding between C410 and C457 (*Lamriben et al., 2018*) might stabilize the MHD of EDEM1 (*Figure 2C*) but might instead decrease the α1,2-mannosidase activity of EDEM1. The resulting M5 has no α1,2-linked mannosyl residue, which in turn means that no other α1,2-mannosidase is necessary for the mammalian gpERAD. In addition, both EDEM3 and EDEM1 nearly equally produced M7A and M7C from M8B (*Figure 5D (a)*), which explains the presence of a small amount of M7C on glycoproteins in HCT116 cells (*Ninagawa et al., 2014*). M7C is likely converted to M6 and then to M5 by EDEM3 and EDEM1, because they have no preference for the mannose residue in the A branch or C branch of M8B. Furthermore, both EDEM3 and EDEM1 efficiently trimmed mannose from M8B on a glycoprotein, such as mCD3-δ-ΔTM-HA (*Figure 6*) and ATF6α(C) (*Figure 7*). Thus, the entire route of oligosaccharide processing and the enzymes responsible for it have been established for mammalian gpERAD for the first time (*Figure 5E*).

It was previously shown that the MHD of EDEM3 is transiently disulfide-bonded to the thioredoxin domain-containing protein ERp46, and that coexpression of EDEM3 and ERp46 enhances EDEM3 activity in gpERAD (*Yu et al., 2018*). Based on our current results, coexpression of EDEM3 and ERp46 is likely to facilitate the formation of disulfide bonding between C82 and C441 of human EDEM3 (C83 and C442 of mouse EDEM3) in the MHD, which is required for EDEM3 to exhibit α1,2-mannosidase activity (*Figure 2D*). Once fully folded with correct disulfide bonding in the MHD, EDEM3 requires no partner protein to exhibit α1,2-mannosidase activity in vitro, in contrast to EDEM2. One question which remains is why only EDEM2 requires TXNDC11 for formation of the disulfide bond in the MHD (*Figure 1D*), even though such disulfide bonding is also essential for the function of EDEM1 and EDEM3 (*Figure 2C and D*).

A second remaining question concerns how the specificity of EDEM2 toward M9 is achieved; although the MHD of EDEM1, EDEM2, and EDEM3 is homologous, EDEM2 effectively converted PA-M9 to PA-M8B (little PA-M9 was present after 24 hr incubation with purified EDEM2-TXNDC11 complex) (*George et al., 2020*), whereas EDEM2 poorly converted M8B to M7 in both PA-bound M8B (*Figure 5C*) and protein-bound M8B (*Figures 6 and 7*), in contrast to EDEM3. We previously showed that the conformation of noncomplexed and complexed EDEM2 with TXNDC11 differs, using sensitivity to trypsin digestion (*George et al., 2020*). It is possible that the structure of the catalytic site of EDEM2 becomes significantly different to that of EDEM1 or EDEM3 following

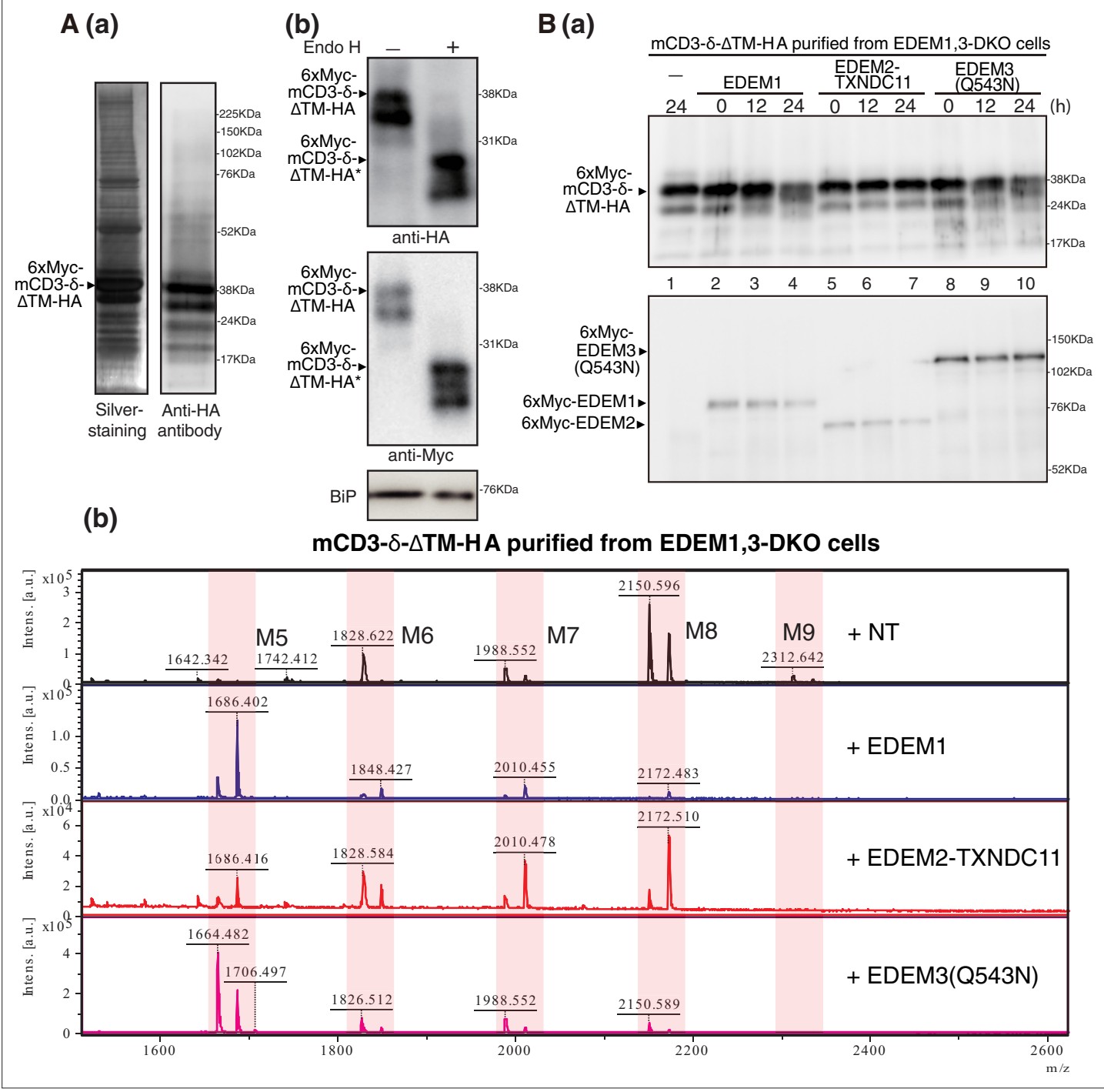

**Figure 6.** Effect of purified EDEM1/2/3 on M8B present in mCD3- δ -ΔTM-HA. (**A**) (**a**) Eluate obtained from EDEM1, 3-double knockout (DKO) cells overexpressing TAP-mCD3- δ -ΔTM-HA was subjected to SDS-PAGE under reducing conditions, silver-stained, and then analyzed by immunoblotting using anti-HA antibody. (**b**) Eluate in (**a**) was untreated (-) or treated (+) with EndoH, subjected to SDS-PAGE under reducing conditions, and analyzed by immunoblotting using anti-HA, anti-Myc, and anti-GRP78 (which is identical to BiP) antibodies. (**B**) (**a**) Eluate in (**A**) was incubated with purified EDEM1, EDEM2-TXNDC11 complex, or EDEM3(Q543N) for the indicated time, and then analyzed by immunoblotting using anti-Myc antibody. (**b**) N-glycans prepared from samples in (**a**) after 24 hr incubation were analyzed by mass spectrometry (MS). This experiment was conducted once.

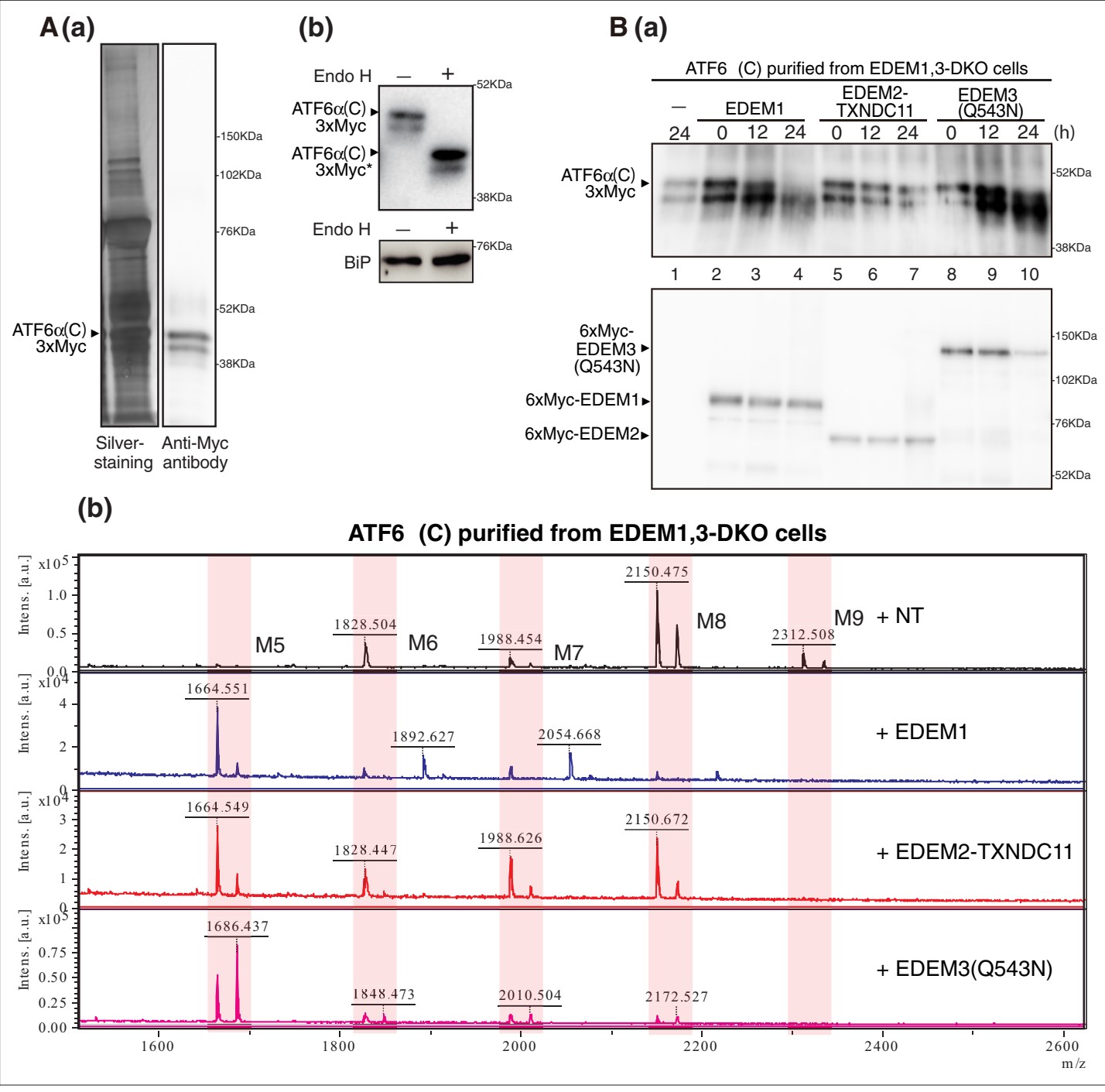

**Figure 7.** Effect of purified EDEM1/2/3 on M8B present in ATF6α(C). (**A**) (**a**) Eluate obtained from EDEM1, 3-double knockout (DKO) cells overexpressing ATF6α(C)-TAP2 was subjected to SDS-PAGE under reducing conditions, silver-stained, and analyzed by immunoblotting using anti-Myc antibody. (**b**) Eluate in (**a**) was untreated (-) or treated (+) with EndoH, subjected to SDS-PAGE under reducing conditions, and analyzed by immunoblotting using anti-Myc and anti-GRP78 (which is identical to BiP) antibodies. (**B**) (**a**) Eluate in (**A**) was incubated with purified EDEM1, EDEM2-TXNDC11 complex, or EDEM3(Q543N) for the indicated time, and then analyzed by immunoblotting using anti-Myc antibody. (**b**) N-glycans prepared from samples in (**a**) after 24 hr incubation were analyzed by mass spectrometry (MS). This experiment was conducted once.

intramolecular disulfide bond formation in the MHD plus intermolecular disulfide bonding between EDEM2 and TXNDC11; this may explain not only the specificity of EDEM2 toward M9 but also the requirement for TXNDC11 as a stable partner of EDEM2. Structural determination is expected to answer these questions.

# Materials and methods

**Key resources table**

| Reagent type (species) or resource | Designation | Source or reference | Identifiers | Additional information |
|---|---|---|---|---|
| Cell line (*Homo sapiens*) | Colorectal carcinoma | ATCC | HCT116 | This cell line has been authenticated and tested negative for mycoplasma. |
| Recombinant DNA reagent | p3xFlag-CMV-14 | Sigma-Aldrich | | |
| Recombinant DNA reagent | pmCherry-N1 | TAKARA | | |
| Recombinant DNA reagent | pSecTag2/Hygro | Thermo Fisher Scientific | | |
| Recombinant DNA reagent | pEGFP-C1 | CLONTECH | | |
| Antibody | Anti-TXNDC11 (rabbit monoclonal) | Abcam | Cat#: ab188329 | WB (1:500) |
| Antibody | Anti-EDEM1 (rabbit polyclonal) | Sigma-Aldrich | Cat#: E8406 | WB (1:500) |
| Antibody | Anti-EDEM2 (rabbit polyclonal) | Novusbio | Cat#: NBP2-37921 | WB (1:500) |
| Antibody | Anti-EDEM3 (mouse monoclonal) | Sigma-Aldrich | Cat#: E0409 | WB (1:500) |
| Antibody | Anti-GRP78 antibody (rabbit polyclonal) | Thermo Fisher Scientific | Cat#: PA1-014A | WB (1:1000) |
| Antibody | Anti-HA (rabbit polyclonal) | Recenttec | Cat#: R4-TP1411100 | WB (1:1000) |
| Antibody | Anti-Flag (mouse monoclonal) | Sigma-Aldrich | Cat#: F3165 | WB (1:1000) IP (2.5 µl) |
| Antibody | Anti-Myc-direct-HRP antibody | MBL | Cat#: M047-7 | WB (1:1000) |

## Statistics

Statistical analysis was conducted using Student's t-test, with probability expressed as $*p < 0.05$ and $**p < 0.01$.

## Construction of plasmids

Recombinant DNA techniques were performed according to standard procedures (*Sambrook et al., 1989*) and the integrity of all constructed plasmids was confirmed by extensive sequencing analyses. Site-directed mutagenesis was carried out using DpnI. A p3xFlag-CMV-14 expression vector (Sigma-Aldrich) was used to express proteins (EDEM1, EDEM2, EDEM3, MAN1B1, and TXNDC11) tagged with 3xFlag at the C-terminus. pCMV-SP-TAP-EDEM2 (*George et al., 2020*) was utilized to construct pCMV-SP-TAP-EDEM1, pCMV-SP-TAP-EDEM3, and pCMV-SP-TAP-MAN1B1. pcDNA3.1-SP-TAP-mCD3-δ-ΔTM-HA was constructed using pcDNA3.1-mCD3-δ-ΔTM-HA. ATF6α(C)-TAP2 containing 3xMyc, TEV protease recognition site, and 2 × immunoglobulin G-binding site of protein A was constructed previously (*Sato et al., 2011*). The ERAD-L substrate mCD3-δ-ΔTM-HA was the kind gift of Maurizio Molinari at the Institute for Research in Biomedicine, Switzerland.

A pmCherry-N1 expression vector (TAKARA) was used to express MAN1B1 fused with mCherry at the C-terminus. To truncate the CMV promoter of pmCherry-N1, the vector was digested with AaII and then self-ligated to create pCMVshort-mCherry-N1. The full-length MAN1B1 with GGGGSGGGGS flexible linker was inserted into the BglII-SalI sites of the pmCherry-N1 or pCMVshort-mCherry-N1 vector using primers 5'-GATAGATCTTGCGATGGCTGCCTGCGAGGGCAG-3' and 5'-ACCGTCGACCCTGAGCCTCCGCCTCCTGAGCCTCCGCCTCCGGCAGGGGTCCAGATAGGC-3'.

To create a construct to express mEGFP-KDEL, EGFP with a monomeric mutation A206K (*Zacharias et al., 2002*) fused with the ER retention signal KDEL was inserted into the HindIII-XhoI sites of the pSecTag2/Hygro vector (Thermo Fisher Scientific). To create a construct to express mEGFP-Giantin, the C-terminal fragment of Giantin responsible for its Golgi localization was subcloned from pmScarlet_Giantin_C1 (*Bindels et al., 2017*), which was obtained from Dorus Gadella (Addgene

plasmid #85048), and inserted into the XhoI-BamHI sites of the pmEGFP-C1 vector, which was generated by introducing the A206K mutation in the EGFP coding sequence of the pEGFP-C1 vector (CLONTECH). The pLamp1-mGFP vector (*Falcón-Pérez et al., 2005*), in which L221 of EGFP was replaced by lysine residue to avoid undesirable dimerization, was obtained from Esteban Dell'Angelica (Addgene plasmid #34831).

## Cell culture, transfection, and *N*-glycan profiling

HCT116 cells (ATCC CCL-247) and HeLa cells were cultured in Dulbecco's modified Eagle's medium (glucose 4.5 g/l) supplemented with 10 % fetal bovine serum, 2 mM glutamine, and antibiotics (100 U/ml penicillin and 100 μg/ml streptomycin) at 37 °C in a humidified 5 % $CO_2$/95 % air atmosphere. Transfection was performed using polyethylenimine max (Polyscience) according to the manufacturer's instructions. Structural identification of *N*-glycans released from total cellular glycoproteins by hydrazinolysis was performed as described previously (*Horimoto et al., 2013*; *Ninagawa et al., 2014*). PA oligosaccharides were first fractionated by HPLC on a TSK-gel Amide-80 (amide-silica) column (Tosoh) and then on a Shim-pack HRC-octadecyl silica column (Shimadzu).

## Live-cell imaging

Cells were plated on 35 mm multi-well glass bottom dishes (Matsunami Glass, D141400). Transfection was performed with X-tremeGENE 9 (Roche) according to the manufacturer's instructions; 16–20 hr after transfection, the growth medium was replaced with phenol red-free Dulbecco's modified Eagle's medium (glucose 4.5 g/l) supplemented with 10 % fetal bovine serum and 2 mM glutamine. During image acquisition, cells were incubated on a Tokai Hit stage top incubator at 37 °C in a humidified 5 % $CO_2$/95 % air atmosphere. Images were acquired with an LSM880 confocal microscope with Airyscan (Carl Zeiss) equipped with a Plan-Apochromat 63 ×/1.4 Oil DIC M27 objective lens. Image acquisition was performed using ZEN software (black edition 2.3). Super-resolution images were obtained by subjecting raw images to the Airyscan processing program. Fiji software was used for image presentation.

## Immunological techniques

Immunoblotting analysis was carried out according to the standard procedure (*Sambrook et al., 1989*) as described previously (*Ninagawa et al., 2011*). Chemiluminescence obtained using Western Blotting Luminol Reagent (Santa Cruz Biotechnology) was detected using an LAS-3000mini Lumino-Image analyzer (Fuji Film).

Immunoprecipitation was performed using anti-Flag antibody and protein G-coupled Sepharose beads (GE Healthcare). Beads were washed with high salt buffer (50 mM Tris/Cl, pH 8.0, containing 1% NP-40 and 150 mM NaCl) twice, washed with PBS, and boiled in Laemmli's sample buffer.

## Asp-N digestion and LC/MS

Samples were dissolved in 40 μl PBS and treated with 0.2 μg of Asp-N (Endoproteinase Asp-N sequence grade, Roche) at 37 °C for 16 hr. The peptides in the sample solution were separated by nano-flow LC using UltiMate 3000 RSLCnano LC system (Thermo Fisher Scientific). The analytical column was a reversed-phase column (PepMap RSLC, C18, Thermo Fisher Scientific; 3 μm, 0.075 mm × 150 mm). The mobile phases A and B were distilled water containing 0.1 % formic acid and acetonitrile containing 0.1 % formic acid, respectively. The flow rate was set to 300 nl/min, and the gradient condition was an isocratic flow at 2 % B for 3 min and a linear gradient from 2 % B to 40 % B for 150 min. The eluted solution was automatically subjected to hybrid ion trap-Orbitrap MS (Orbitrap fusion Lumos mass spectrometer, Thermo Fisher Scientific). Data-dependent EThcD-tandem MS (EThcD-MS/MS) and data-dependent HCD-MS/MS/MS of product ions generated in the data-dependent EThcD-MS/MS were performed in the linear ion trap. The MS conditions were as follows: electrospray voltage, 2.0 kV in positive ion mode; capillary temperature, 250 °C; full mass resolution, 120,000; full mass range, *m/z* 500–1500; isolation window width for data-dependent scan, 2; maximum injection time for data-dependent EThcD-MS/MS, 100 ms; and collision energy for data-dependent HCD-MS/MS/MS, 35 %. To identify disulfide-linked peptide from EDEM2, EThcD-MS/MS data were subjected to database search analysis using the BioPharma Finder 3.0 software (Thermo Fisher Scientific), with peptide mass tolerance set to ±5 ppm. The amino sequencing of disulfide-linked peptides

was also manually performed using HCD-MS/MS/MS spectra of disulfide-dissociated peptide ions generated by EThcD-MS/MS.

## Genomic PCR

Homologous recombination in HCT116 cells was confirmed by genomic PCR using a pair of primers 5'-CTATGTGCCAGCTACCATGTG-3' and 5'-TACTCCATGGAGGCCAAGCC-3' for hEDEM1, or 5'-GAGTACAGAGAGAAAAAGGAC-3' and 5'-GCCACTAGTCTCCATCGCGC-3' for hEDEM3.

## RT-PCR

Total RNA prepared from cultured cells (~3 × 10$^6$ cells) by the acid guanidinium/phenol/chloroform method using ISOGEN (Nippon Gene) was converted to cDNA using Moloney murine leukemia virus reverse transcription (Invitrogen) and random primers. The full-length open reading frame of EDEM1, EDEM2, or EDEM3 was amplified using PrimeSTAR HS DNA polymerase (Takara Bio) and a pair of primers described previously (*Ninagawa et al., 2014*).

## Purification of EDEM1, EDEM3, and MAN1B1

EDEM2-TXNDC11 complex was purified as described previously (*George et al., 2020*). WT HCT116 cells plated on 15 cm dishes were transfected with plasmid to express TAP-tagged EDEM1, EDEM3(Q543N), or MAN1B1(Δ105). Forty-eight hours later, the cells were lysed in lysis buffer (50 mM MES, pH 7.5, containing 150 mM NaCl, 1 % CHAPS, and EDTA-free protease inhibitor cocktail [Roche]), and centrifuged at 9500 *g* at 4 °C for 30 min. The resulting supernatant was filtrated through a low protein binding syringe filter (Merck) and rotated at 4 °C for 3 hr after the addition of IgG Sepharose beads (GE Healthcare). The beads were collected by centrifugation at 3000 rpm at 4 °C for 1 min, washed twice briefly and then washed at 4 °C overnight with wash buffer (50 mM MES, pH 7.5, containing 400 mM NaCl, 0.1 % CHAPS, and EDTA-free protease inhibitor cocktail). The beads were then incubated with 200 U of AcTEV protease (Invitrogen) in TEV buffer (50 mM MES, pH 7.5, containing 150 mM NaCl) at 4 °C for 24 hr, and then centrifuged briefly. The resulting supernatant was concentrated using an Amicon Filter (10 kDa cutoff, Millipore) by centrifugation at 4000 *g* at 4 °C for 1 hr. During concentration, the buffer was changed to 50 mM MES, pH 7.5, containing 150 mM NaCl and 5 mM CaCl$_2$, by three additions to the filter.

## In vitro mannosidase assay

PA-labeled free oligosaccharides were purchased from Takara Bio. Approximately 1.0 pmol of purified EDEM1, EDEM2-TXNDC11 complex, EDEM3(Q543N), or MAN1B1(Δ105) was incubated with 150 pmol PA-M8B in a total volume of 45 µl of assay buffer (50 mM MES, pH 7.5, containing 150 mM NaCl, and 5 mM CaCl$_2$) at 37 °C for 24 hr. The reaction was stopped by boiling for 5 min. The samples were evaporated, dissolved in 20 µl of 70 % (v/v) acetonitrile, and analyzed using a TSK-gel Amide-80 column (Tosoh) for mannose contents. Identification of *N*-glycan structures was based on their elution positions on the column and their molecular mass values compared with those of PA-glycans in the GALAXY database (http://www.glycoanalysis.info/galaxy2/ENG/index.jsp) (*Takahashi and Kato, 2003*). The peaks of M7, M6, and M5 were collected, evaporated, dissolved in 20 µl of water, and analyzed using a Shim-pack HRC-octadecyl silica column (Shimadzu) for isomer identification.

## Purification of mCD3-Δ-ΔTM-HA and ATF6α(C)

EDEM1, 3-DKO cells plated on 15 cm dishes were transfected with plasmid to express TAP-tagged mCD3-δ-ΔTM-HA or ATF6α(C) tagged with TAP2. 6xMyc-tagged mCD3-δ-ΔTM-HA and ATF6α(C) tagged with 3xMyc were purified as described above for α1,2-mannosidases.

## *N*-glycan analysis by MALDI-TOF MS

Approximately 800 ng of gpERAD substrate (6xMyc-mCD3-δ-ΔTM-HA or ATF6α(C)–3xMyc) reacted with 100–150 ng (depending on molecular weight) of α1,2-mannosidase at 37 °C for 24 hr were treated with 500 units of PNGase (New England Biolabs) at 37 °C overnight. The released *N*-glycans were captured and labeled with aminooxy-functionalized tryptophanyl arginine methyl ester (aoWR) by BlotGlyco beads (Sumitomo Bakelite) as described previously (*Furukawa et al., 2008*; *Uematsu et al., 2005*). Matrix-assisted laser desorption/ionization-time of flight mass spectrometry (MALDI-TOF

MS) analyses of aoWR-labeled glycans were performed on Autoflex Speed (Bruker Daltonics) operated in positive-ion reflector mode. For MS acquisition, aoWR-labeled glycans in acetonitrile were mixed 1:1 with dihydrobenzoic acid (10 mg/ml in 50 % acetonitrile) and spotted on the target plate.

## Acknowledgements

The authors declare no competing financial interests. We thank Ms Kaoru Miyagawa for her technical and secretarial assistance. This work was financially supported in part by grants from MEXT, Japan (18K06216 to SN, 17H06414 and 21H02625 to HY, 19K06658 to TI, 20K21495 to KK, 18K06110 to TO, 17H01432 and 17H06419 to KM), the Takeda Science Foundation and Kobayashi Foundation (to SN), and by the Joint Research by Exploratory Research Center on Life and Living Systems (19–303 to KM, 20–305 to KM, and 21–307 to SN). We are grateful to the Analytical Center, Meijo University, for the analysis of oligosaccharides using MALDI-TOF/MS, Autoflex Speed (Bruker).

## Additional information

### Funding

| Funder | Grant reference number | Author |
|---|---|---|
| Ministry of Education, Culture, Sports, Science and Technology | 18K06216 | Satoshi Ninagawa |
| Ministry of Education, Culture, Sports, Science and Technology | 17H06414 | Hirokazu Yagi |
| Ministry of Education, Culture, Sports, Science and Technology | 21H02625 | Hirokazu Yagi |
| Ministry of Education, Culture, Sports, Science and Technology | 19K06658 | Tokiro Ishikawa |
| Ministry of Education, Culture, Sports, Science and Technology | 20K21495 | Koichi Kato |
| Ministry of Education, Culture, Sports, Science and Technology | 18K06110 | Tetsuya Okada |
| Ministry of Education, Culture, Sports, Science and Technology | 17H01432 | Kazutoshi Mori |
| Ministry of Education, Culture, Sports, Science and Technology | 17H06419 | Kazutoshi Mori |
| Takeda Science Foundation | | Satoshi Ninagawa |
| Kobayashi Foundation | | Satoshi Ninagawa |
| Joint Research by Exploratory Research Center on Life and Living Systems | 19–303 | Kazutoshi Mori |
| Joint Research by Exploratory Research Center on Life and Living Systems | 20–305 | Kazutoshi Mori |

| Funder | Grant reference number | Author |
|---|---|---|
| Joint Research by Exploratory Research Center on Life and Living Systems | 21–307 | Satoshi Ninagawa |

The funders had no role in study design, data collection and interpretation, or the decision to submit the work for publication.

## Author contributions

Ginto George, Satoshi Ninagawa, Hirokazu Yagi, Jun-ichi Furukawa, Noritaka Hashii, Akiko Ishii-Watabe, Ying Deng, Kazutoshi Matsushita, Tokiro Ishikawa, Yugoviandi P Mamahit, Yuta Maki, Yasuhiro Kajihara, Koichi Kato, Tetsuya Okada, Investigation; Kazutoshi Mori, Conceptualization, Supervision, Writing – original draft

## Author ORCIDs

Ginto George http://orcid.org/0000-0003-4804-9594
Satoshi Ninagawa http://orcid.org/0000-0002-8005-4716
Hirokazu Yagi http://orcid.org/0000-0001-9296-0225
Jun-ichi Furukawa http://orcid.org/0000-0002-7284-2261
Noritaka Hashii http://orcid.org/0000-0003-1363-2050
Ying Deng http://orcid.org/0000-0003-4657-9284
Kazutoshi Matsushita http://orcid.org/0000-0002-5666-9630
Tokiro Ishikawa http://orcid.org/0000-0003-1718-6764
Yugoviandi P Mamahit http://orcid.org/0000-0002-2974-7795
Yuta Maki http://orcid.org/0000-0002-5838-302X
Yasuhiro Kajihara http://orcid.org/0000-0002-6656-2394
Koichi Kato http://orcid.org/0000-0001-7187-9612
Tetsuya Okada http://orcid.org/0000-0002-2513-1301
Kazutoshi Mori http://orcid.org/0000-0001-7378-4019

## Decision letter and Author response

Decision letter https://doi.org/10.7554/eLife.70357.sa1
Author response https://doi.org/10.7554/eLife.70357.sa2

# Additional files

## Supplementary files

• Transparent reporting form

## Data availability

All data generated or analyzed during this study are included in the manuscript and supporting files. Source data files have been provided for Figures 1, 1S1, 2, 2S1, 4, 5, 6 and 7.

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
