## [Editor Report]

This study demonstrates the details of mannose trimming in the ER by EDEM1 and EDEM3, filling an important gap in how mannose trimming creates an ERAD signal and shunts glycoproteins to the ERAD pathway.

---

## [Decision Letter]

**Decision letter after peer review:**

Thank you for submitting your article "Purified EDEM3 or EDEM1 Alone Produces Determinant Oligosaccharide Structures from M8B in Mammalian Glycoprotein ERAD" for consideration by *eLife*. Your article has been reviewed by 2 peer reviewers, and the evaluation has been overseen by a Reviewing Editor and Vivek Malhotra as the Senior Editor. The following individual involved in review of your submission has agreed to reveal their identity: Tadashi Suzuki (Reviewer #2).

Essential revisions:

N-glycans act as sorting signals in the secretory pathway with mannose trimming marking proteins for turnover through the endoplasmic reticulum associated degradation (ERAD) pathway by exposing the alpha1,6-linked mannose residues on the C-antennary of the N-glycan. The ER possesses a number of mannosidase or mannosidase-like family members including EDEM1-3. Previous work by the authors identified EDEM2-TXNDC11 as catalyzing the initial trimming step from Man9 to Man8B (George et al., *eLife* 2020). Now they show that EDEM1 and EDEM3 further trim Man8B thereby creating the alpha1,6-linked mannose signal. Thus, the work fills an important gap in our understanding of the mannose trimming pathway that creates an ERAD signal for sorting of glycoproteins to the ERAD pathway.

The expert reviewers were, on the whole, convinced by the work. Nevertheless, there were two important issues that arose. The first two are summarized here and then the full comments by both reviewers can be found below. Please submit a revision along with a cover letter that explains how all these issues were addressed in the revised version.

1. The first concerns the conclusion that EDEM1 and EDEM3 are THE mannosidases responsible for trimming from M8B to M5-7 in the ER. If the authors standby the conclusion that these enzymes are both sufficient (as shown here) and also required (in vivo) they should clearly explain the data supporting this in the text.

2. In its current form the study is unclear about the selectivity of the enzymes towards their targets. Are they preferentially acting on partially folded/misfolded substrates? The authors should clarify this with additional data if necessary. An attempt should be made to broaden the Discussion a bit for the general reader to place the activity of these enzymes into the cellular process that selects proteins for ERAD.

*Reviewer #1:*

This study describes the processing of glycans by members of the EDEM family in preparation for ERAD. N-glycans acts as sorting signals in the secretory pathway. Mannose trimming marks proteins for turnover through the endoplasmic reticulum associated degradation (ERAD) pathway by exposing the alpha1,6-linked mannose residues on the C-antennary of the N-glycan. The ER possesses a number of mannosidase or mannosidase-like family members including EDEM1-3. The Mori lab has previously used knockout human cell lines to explore the order that the mannosidases act on the glycan (Ninagawa et al., JCB 2014). They also showed that EDEM2-TXNDC11 performs the initial trimming step from Man9 to Man8B (George et al., *eLife* 2020). The current study provides a continuation of this study that provides biochemical support that EDEM2 trimming is followed by EDEM1 and EDEM3 exposing the key alpha1,6-linked mannose ERAD signal (as well as performing further trimming). EDEM1 and EDEM3 are shown to possess a key large conserved intramolecular disulfide in their mannosidase homology domain, as they have seen previously for EDEM2 (George et al., *eLife* 2020). This intramolecular disulfide is required for their mannosidase activity. The most significant finding is the demonstration that purified EDEM1 and EDEM3 further trim Man8B (created by EDEM2-TXNDC11). This was shown both for isolated glycans (pyridylamine labeled M8B, Figure 4) and glycans linked to ERAD substrates (mCD3 (Figure 5); and ATF6alpha (Figure 6)). Overall, this study demonstrates the details of mannose trimming in the ER by EDEM1 and EDEM3, filling an important gap in how mannose trimming creates an ERAD signal and shunts glycoproteins to the ERAD pathway.

The major strength of the study is the identification of the mannosidase activities of EDEM1 and EDEM3. They show they these two ER mannosidases are responsibility for generating the ERAD signal, Man7C using glycomics and purified components. While this has been hypothesized, direct demonstration of the mannosidase activity with glycomics has been missing.

Figure 2C involves the isolation of total cellular glycosidase (Pngase F?) liberated glycans (details on the experimental procedure are sparse). As EDEMs are expected to act specifically on non-native clients for ERAD targeting, wouldn't most glycans be expected to be inert to the presence of the EDEMs? Or do the EDEMs act on native substrates too? Would mannose trimming be accentuated by chemical folding disruptors? Or can glycans on ERAD substrates specifically be analyzed?

The EDEMs are expected to have some querying abilities to help mark non-native proteins rather than acting as promiscuous mannosidases, yet this is not discussed in the manuscript. Is it due to localization within ER subdomains or the selectivity of the mannosidases?

Why isn't there mannosidase trimming in WT Figure 4C? Shouldn't there be endogenous mannosidase activity in the WT cells?

*Reviewer #2:*

This manuscript provides a biochemical evidence intramolecular formation of S-S bond in EDEM2, which is possibly catalyzed by TXNDC11, a binding partner for EDEM2. Then they also characterize the activity of EDEM1 and EDEM3, and showed that both EDEM3 and EDEM1 is capable of processing oligosaccharides to expose alpha1,6-mannose, a key residue for recognition by OS-9 lectins.

While it has been assumed that both EDEM1 and 3 have an alpha1,2-mannosidase activity, whether there are additional factor(s), such as oxidoreductase, required for its enzyme activity or not remain unclarified. This reviewer therefore likes their biochemical approaches to show that, at least for EDEM3, it looks like that the S-S bond formation with another protein is not likely to be involved in its enzymatic activity (Figure 3A(b)). They also clearly showed that EDEM1 or EDEM3 can exhibit alpha1,2-mannosidase activity in vitro, towards both PA-labeled glycans and N-glycans on ERAD model substrates.

The reviewer feels, however, that their conclusion that EDEM1 and EDEM3 is THE mannosidase responsible for trimming from M8B to M5-7 in the ER are not sufficiently supported. It is clear that their activities in vitro do not necessarily lead to this conclusion (as MAN1B1 can also clearly exert the similar activity in vitro (ex Figure 4C)). Their conclusion, the reviewer believes, is mainly coming from their previous observation that, MAN1B1-KO did not result in drastic change of N-glycan isomers of total cellular glycoproteins (similar analysis with Figure 2C of current study).

While the reviewer acknowledges that the authors mentioned about that in the text (Page 4, lines 15-16), as the current form their logic on drawing conclusion of Figure 4E is not clear. The reviewer thus urges the authors to clearly spell out this point in the text (i.e. why they think Man1B1 is not involved in this process). Otherwise the readers who are not familiar with the authors' previous studies may not understand their logic.

---

## [Author Response]

Essential revisions:N-glycans act as sorting signals in the secretory pathway with mannose trimming marking proteins for turnover through the endoplasmic reticulum associated degradation (ERAD) pathway by exposing the alpha1,6-linked mannose residues on the C-antennary of the N-glycan. The ER possesses a number of mannosidase or mannosidase-like family members including EDEM1-3. Previous work by the authors identified EDEM2-TXNDC11 as catalyzing the initial trimming step from Man9 to Man8B (George et al., eLife 2020). Now they show that EDEM1 and EDEM3 further trim Man8B thereby creating the alpha1,6-linked mannose signal. Thus, the work fills an important gap in our understanding of the mannose trimming pathway that creates an ERAD signal for sorting of glycoproteins to the ERAD pathway.The expert reviewers were, on the whole, convinced by the work. Nevertheless, there were two important issues that arose. The first two are summarized here and then the full comments by both reviewers can be found below. Please submit a revision along with a cover letter that explains how all these issues were addressed in the revised version.1. The first concerns the conclusion that EDEM1 and EDEM3 are THE mannosidases responsible for trimming from M8B to M5-7 in the ER. If the authors standby the conclusion that these enzymes are both sufficient (as shown here) and also required (in vivo) they should clearly explain the data supporting this in the text.

We have shown that MAN1B1 fused to mCherry is localized in the Golgi apparatus but not in the ER in both HCT116 cells (Figure 3) and HeLa cells (Figure 3-S1) (text: p. 9, line 14 – p. 10, line 13). We have added “Our current results with another a1,2-mannosidase MAN1B1 (Figure 3 and Figure 3-S1) are well consistent with a previous report showing the Golgi localization of endogenous MAN1B1 by immunofluorescence (Pan et al., 2011). […] The system becomes effective if all components of gpERAD exist in the same compartment, namely in the ER” (p. 16, lines 11-21).

2. In its current form the study is unclear about the selectivity of the enzymes towards their targets. Are they preferentially acting on partially folded/misfolded substrates? The authors should clarify this with additional data if necessary. An attempt should be made to broaden the Discussion a bit for the general reader to place the activity of these enzymes into the cellular process that selects proteins for ERAD.

We submitted this manuscript as a Research Advance of our previous *eLife* paper in 2020, in which the mannosidase activity of the EDEM2-TXNDC11 complex was demonstrated using PA-M9 as a substrate. The Author Guide says that “Where appropriate, the Research Advance will be considered by the same editors and reviewers who were involved in the assessment of the original paper”. However, reviewer #1 was changed from Dan Hebert to another person, and we feel that the assessment criteria have changed considerably.

In this manuscript we demonstrated the mannosidase activity of EDEM3 and EDEM1 using not only PA-M8B but also two glycoproteins as a substrate, which should be sufficient for publication, as in the case of our *eLife* paper 2020.

Indeed, reviewer #2 liked our biochemical approaches to show that, at least for EDEM3, it looks like that the S-S bond formation with another protein is not likely to be involved in its enzymatic activity. This is an important finding, and therefore we use the title “Purified EDEM3 or EDEM1 Alone Produces Determinant Oligosaccharide Structures from M8B in Mammalian Glycoprotein ERAD”.

Because the selectivity of the enzymes towards their targets is most critical to the EDEM2-TXNDC11 complex, which initiates glycoprotein ERAD, we firmly consider that this criticism 2 is beyond the scope of this manuscript, which mainly focused on EDEM3 and EDEM1.

Reviewer #1:[…] Figure 2C involves the isolation of total cellular glycosidase (Pngase F?) liberated glycans (details on the experimental procedure are sparse).

We have described that “Structural identification of *N*-glycans released from total cellular glycoproteins by hydrazinolysis was performed as described previously (Horimoto et al., 2013; Ninagawa et al., 2014) (p. 22, lines 4-6).

As EDEMs are expected to act specifically on non-native clients for ERAD targeting, wouldn't most glycans be expected to be inert to the presence of the EDEMs? Or do the EDEMs act on native substrates too? Would mannose trimming be accentuated by chemical folding disruptors? Or can glycans on ERAD substrates specifically be analyzed?The EDEMs are expected to have some querying abilities to help mark non-native proteins rather than acting as promiscuous mannosidases, yet this is not discussed in the manuscript. Is it due to localization within ER subdomains or the selectivity of the mannosidases?

Please refer to our response describe above.

Why isn't there mannosidase trimming in WT Figure 4C? Shouldn't there be endogenous mannosidase activity in the WT cells?

We have changed (WT HCT116) to untransfected (new Figure 5C). Essentially no cellular proteins were recovered from untransfected HCT116 cells after TAP-mediated purification (lane 1 from the right in new Figure 5B).

Reviewer #2:[…] The reviewer feels, however, that their conclusion that EDEM1 and EDEM3 is THE mannosidase responsible for trimming from M8B to M5-7 in the ER are not sufficiently supported. It is clear that their activities in vitro do not necessarily lead to this conclusion (as MAN1B1 can also clearly exert the similar activity in vitro (ex Figure 4C)). Their conclusion, the reviewer believes, is mainly coming from their previous observation that, MAN1B1-KO did not result in drastic change of N-glycan isomers of total cellular glycoproteins (similar analysis with Figure 2C of current study).While the reviewer acknowledges that the authors mentioned about that in the text (Page 4, lines 15-16), as the current form their logic on drawing conclusion of Figure 4E is not clear. The reviewer thus urges the authors to clearly spell out this point in the text (i.e. why they think Man1B1 is not involved in this process). Otherwise the readers who are not familiar with the authors' previous studies may not understand their logic.

Please refer to our response describe above.